# Complied by Belief Consistency: The Cognitive-Information Lens of User-Generated Persuasion

**Hung-Pin Shih [1],\*, Kee-hung Lai [2]** and **T. C. E. Cheng [2]**

[1] School of Business, Minnan Science and Technology University, Quanzhou 362332, China
[2] Faculty of Business, The Hong Kong Polytechnic University, Hong Kong 999077, China
\* Correspondence: hungpin.shih@gmail.com

**Abstract:** Confirmation biases make consumers feel comfortable because consistent beliefs simplify the processing of electronic word-of-mouth (eWOM). Whether the helpfulness of eWOM is a belief of information underlying *biased information*, i.e., positive–negative asymmetry, or an illusion of overconfidence underlying *biased judgment*, i.e., belief consistency, is crucial to the foundation of theory and the advance of practice in user-generated persuasion. The questions challenge the literature that the helpfulness of product reviews relies on unbiased information and/or unbiased judgment. Drawing on the cognitive-information lens, we developed a research model to explain how *belief consistency* affects the helpfulness beliefs of eWOM, and examined the effects of positive–negative asymmetry. Using a scenario-based questionnaire survey, we collected 334 consumer samples to test the research model. According to the empirical results, the conflicts of influence between positive and negative confirmation indicated that perceived review helpfulness was a belief of information and constrained by the positive–negative review frame. Without using personal expertise, respondents' consistent beliefs were significant to confirm positive reviews as useful and thereby perceive the review content as helpful, which is an illusion of overconfidence and constrained by belief consistency. Whether personal expertise reinforces the effect of belief consistency depends on the positive–negative asymmetry.

**Keywords:** confirmation biases; cognitive-information lens; belief consistency; positive-negative asymmetry; helpfulness beliefs

## 1. Introduction

Electronic word-of-mouth (eWOM) that stems from user-generated persuasion of personal experiences in consumption can be used to foster a consumer-dominated channel of marketing communication [1]. Helpful eWOM not only encourages posters' contributions, but also facilitates consumers to determine personal choices [2], and ultimately reinforces both parties' belief systems according to comfortable feelings. Belief systems cause people to develop narratives to express what they believe and define their sense of reality [3]. Review helpfulness is defined as the extent to which review information can help consumers undertake purchase decisions that need credible and diagnostic information [4]. Despite other views, we follow the definition of review helpfulness concerning the valuation of review information to satisfy personal choices. Perceived review helpfulness does not completely reflect the information about product quality, but only exhibits the degree of self-confidence of the readers' beliefs of the review information, to undertake better purchase decisions. However, the spread of biased opinions about products may encourage the manipulation of reviews to mislead naive consumers to skew market competition [5]. Despite the concern of potential manipulation, the helpfulness beliefs of eWOM, based on either biased or unbiased review information, are easily distorted by readers' judgment frames, anchoring points, availability of heuristics, and different psychological perspectives [6–8] or false belief construction that may foster the spread of misinformation [9].

The route people choose determines where they are; the medium people access determines who is connected; and the information people put into their belief systems determines *why* and *how* they are biased toward the judgment. People such as social elites often believe the reason for personal business success, and attribute their success to sufficient effort rather than a luck factor [10]. Intuitive thinking, or the availability of heuristic, probably shapes people's personal choices [6,7]. Other than being by chance, the phenomenon is better understood by the availability of associations leading people to reflect self-enhancement biases under the constraints of incomplete information or limited personal ability. Self-enhancement bias refers to the tendency to endorse self-views that are more favorable than counter-views of objective reality [11]. The assessment of user-generated content resembles an error-prone cognitive process, because people's overconfidence in prior beliefs encourage the search for more consistent reviews to conform to their beliefs [12]. Overconfidence refers to an optimistic estimate of the likelihood of a favorable outcome [13]. To explain how people process incomplete information and why they adopt *biased information*, i.e., *the collection of consistent information or one-sided content in information processing*, we consider if the examination of eWOM can provide reasonable explanations of biased judgment, i.e., *judgment based on consistent beliefs*. In this study, we take belief consistency (or consistent beliefs) as the notion of biased judgment. The wisdom-of-crowds is believed as being helpful for collaborative software development, such as open source or crowdsourcing. Similarly, consumers may view eWOM, i.e., one type of wisdom-of-crowds, as helpful information for judging product quality because the average error of individuals is smaller than an individual's average error [14,15]. However, consistent beliefs or the collection of consistent user-generated persuasion instead of product attributes may enlarge the average error of belief systems (e.g., preferences, experiences, utilities, etc.), which needs deeper examination.

Owing to the burden of physical searches and the cognitive efforts of information processing, consumers often take the strategy of least effort (e.g., ask others or read reviews) to evaluate product quality under the constraint of knowledge boundary [16]. Smart consumers need product reviews instead of advertising information about quality evaluations for intended purchases, because two-sided user-generated content is more comprehensive than one-sided persuasion [17]. Consumers need answers as to whether a positive, or negative, eWOM is more helpful. Considering the tendency of firms to manipulate the spread of positive rather than negative reviews, consumers tend to consider negative information as more diagnostic and, thus, helpful for product evaluations, namely, a negativity bias [18]. In contrast, practical follow-up surveys of customer feedback from product categories, e.g., books, magazines, DVDs, videos, flowers, and food, considered that positive reviews were more influential than product attributes as the clue to predicting repurchasing intention, namely, a positivity bias [19]. Satisfied customers are more likely to spread positive product reviews in contrast with dissatisfied shoppers, who tend to spread negative opinions [20]. The phenomenon that users perceive positive or negative information as having more weight or impact on their impressions, i.e., that the information content is perceived as either more positive, less negative or less positive, or more negative, is termed as *positive–negative asymmetry* [21]. We consider positive–negative asymmetry, i.e., either positivity bias or negativity bias, as the notion of *biased information* that may change the anchoring point of initial beliefs [8]. Either positivity bias or negativity bias is crucial to the helpful evaluation of product reviews [22]. The positive–negative asymmetry of user-generated content may result in the collection of biased (or one-sided) review information to evaluate review helpfulness [23].

Novice consumers probably make purchase decisions by relying on others' opinions about product attributes. Hence, the evaluation of review attributes to judge helpful eWOM seems reasonable for consumers [4,24]. The debate about which review attribute is more relevant to examine product quality or learn consumption experiences and thereby help predict purchase intentions never stops. There exist too few clues to valuing personal choices, given the growing number of user-generated reviews and opinions [25]. The

approach to examine which attributes are desired to determine the helpfulness of a review needs greater effort in text mining, which is beyond our research scope. Owing to limited time and personal effort, most consumers are assumed to judge review helpfulness in terms of a few, rather than all, attributes, implying that the use of specific review attributes can work as an alternative scheme to replace text mining [2]. The helpfulness or unhelpfulness based on the evaluation of review information serves the belief development on the poor–rich spectrum of information values. Previous studies have not yet achieved the use of a common measure to estimate review helpfulness, despite much research being undertaken [2,4]. People adapt to the common measure of "*money*" to estimate product value or the market value of an exchange. The use of a common measure for evaluating review helpfulness, such as helpfulness scores or helpfulness votes, is a stop-gap measure under a variety of human desires and distinct valuation systems. The approaches to *shape* review helpfulness are different from the scheme to *measure* it. The former refers to the influence of review persuasion associated with uncertainty of human judgment [26], which is distorted by biased judgment. The latter refers to the measurement of uncertainty reduction in the judgment [27], which is distorted by biased information. For example, reviewers' profiles can shape users' perceived helpfulness of a review [28], but are unlikely to become a common measure of review helpfulness. We consider whether biased judgment is built on *cognitive processing* and biased information is built on *information processing*, which have been, to date, unexamined in the literature. Building on the cognitive-information lens, in this paper we develop an approach to address *how cognitive processing is built on information processing*. Two assumptions were developed to foster the cognitive-information lens of eWOM evaluations. First, that belief consistency and positive–negative asymmetry jointly shape the processing of eWOM that is reinforced by the self-enhancement of belief systems. Second, that the valuation of eWOM is directly built on positive–negative asymmetry.

The approach to examine product review helpfulness is similar to the approach to find the favor that can satisfy a consumer, or the valence of sentimental messages [29], which is often constrained by the estimation of incomplete information and judgment on subjective preference. The adoption of product reviews instead of quality attributes to predict product quality is reasonable for inexperienced consumers that desire helpful experience information. For example, consumers tend to take source credibility of a review as helpful information for their purchase judgment [2]. However, source credibility is built on social identity and reputation, that appeals to consumers with strong and homophonous ties rather than those in other weak-tie groups [30]. People may not completely reduce the uncertainty of messages or avoid the biased information in opinions. Hence, cognitive biases of human reasoning, such as confirmation biases—a theoretical lens of heuristic processing, would likely distort decision making, leading to the judgment based on personal characteristics rather than complete information or perfect rationality, i.e., an unrealistic assumption that would not exist everywhere. Confirmation bias refers to as an individual's tendency to accept opinions or views that can conform to their pre-existing beliefs, expectations, or hypotheses in cognitive processes [31]. Confirmation biases prompt people to sustain the status quo bias without striving for new evidence or investing more cognitive effort in inconsistent thinking. Hence, confirmation biases foster people to engage in self-enhancement of their belief systems [32]. Confirmation biases can serve as the boundary condition of review assessment [33], but not to directly affect review helpfulness. Hence, overconfidence in the narratives of belief systems fosters self-enhancement of consistent beliefs. Belief consistency is a core element of confirmation bias theory, and defined herein as cognitive consistency between existing beliefs of eWOM and prior beliefs of products in one's belief systems. Given that cognitive processes are constrained by information uncertainty and ambiguity [16] or bounded rationality [34], cognitive consistency fosters one's self-enhancement of pre-existing beliefs, and thereby guides judgment based on consistent beliefs [35,36].

In this study, biased judgment and belief consistency, as well as biased information and positive–negative asymmetry, are used interchangeably, depending on the general or

specific context of user-generated persuasion. Previous studies considered that the helpfulness of product reviews was judged on source- and content-based features [2,17,37,38]. We consider that the "helpfulness beliefs" of product reviews are determined by the effects of biased judgment stemming from recipients and the effects of biased information stemming from review content. Previous research examined the existence of confirmation biases among experts on the valuation of stock-related information [39] or the forecast of business earnings [40]. In contrast, we focus on the presence of belief consistency and its effects on eWOM evaluation. The phenomenon of confirmation biases reinforces the degree of judgment based on consistent beliefs, but the presence of belief consistency does not verify confirmation biases. People need consistent beliefs for effective judgement (e.g., less effort and time), but consistent beliefs enable them to ignore invisible constraints on their judgment. To the best of our knowledge, this study is the first to examine the effects of belief consistency on the evaluation of review helpfulness. Personal expertise is supposed to make sense of the evaluation of information usefulness [41,42], and is thereby examined in the study. Considering that biased judgment or biased information may distort helpfulness beliefs, we address three research questions (*RQs*) concerning consumers' judgment on user-generated persuasion. They are:

*RQ1*: How the positive–negative asymmetry of eWOM contributes to helpfulness beliefs.
*RQ2*: How belief consistency determines the helpfulness beliefs of eWOM.
*RQ3*: Whether personal expertise fosters or inhibits the effects of belief consistency.

The rest of the paper is organized as follows. In Section 2, we explain the theoretical foundation of this study and propose the research model and relevant hypotheses. In Section 3, we describe our methodology for sample data collection and the development of measurement items. We present our empirical results of reliability and validity tests, and hypothesis tests, in Section 4. In Section 5, we discuss our key findings, as well as theoretical and practical implications, according to the findings. We draw conclusions and also present limitations of this study in Section 6.

## 2. Theoretical Foundation

### 2.1. The Cognitive-Information Lens

Cognitive inconsistency causes people to feel psychologically uncomfortable and, thus, resist seeking information that may increase the dissonance of cognitions [43]. Recent studies have widely applied cognitive dissonance theory to address how people's belief systems work, such as customer participation in service recovery [44], the order of evidence presentation in criminal law trials [45], consumer attitudes towards counterfeiting and purchase intentions [46], and employees' compliance with collective beliefs regardless of individuals' belief systems [47]. This study does not address the conflicts of cognitive dissonance—whether to change the behavior so as to avoid regretful conduct, or the cognition for avoiding psychologically uncomfortable feelings. According to Wason [48,49], people typically seek consistent rather than inconsistent instances or information when they test a hypothesized rule. Probably, disconfirmation of a hypothesis would appear as making less contribution, or being unacceptable for scientists. Confirmation biases explain a general phenomenon in that an individual's preliminary hypotheses become a dominant advantage in the judgment process even if the hypotheses are often false [50]. From psychological research, confirmation biases lead people to seek confirmatory evidence to support their existing beliefs, expectations, or hypotheses, rather than counter-argument, counter-evidence, or exception data [31]. People tend to make decisions by searching for information that can conform to their prior beliefs, namely, *selection bias*, or engaging in the interpretation of ambiguous information that can enhance their confidence in prior beliefs, namely, *biased interpretation* [39]. For example, domain experts also exhibit confirmation biases in which they advise people on real-world decision making [51]. Investors often seek stock-related messages (e.g., news or dreams) to confirm, rather than be against, their existing beliefs in the prediction of stock prices [39]. The confirmation biases would produce a twofold effect on personal choices that rely more on the preferred paths of cognitive

processing—the first is *overconfidence, i.e., a higher-than-warranted degree of certainty, on existing consistent beliefs,* and the second is *resistance to making a judgment based on inconsistent beliefs* [31]. In summary, confirmation biases often cause over-optimistic expectations about the probability of personal choices [52]. In the latest example, some people have underestimated the health risks during the COVID-19 pandemic [53].

Researchers have distinct views and inferences about the causes of confirmation bias [54–57]. According to Baddeley [58], people often make a judgment by anchoring a reference point stored in one's pre-existing beliefs, leading to judgment on the heuristics that seek to confirm the beliefs. Hence, anchoring and adjustment heuristics foster people's belief development to follow certain paths in prior experiences. People rely more on the habitual way of acting without assessing better alternatives to re-examine their existing beliefs according to the current situation or the activation of goal pursuit [59]. The habitual way of cognitive processing helps people reduce information uncertainty by dropping the messages that go against their pre-existing beliefs, or using available experiences [60,61]. We consider individual path dependency as a better reason for explaining the phenomenon of confirmation biases [62]. Owing to path dependency, prior experiences in technology use determine the user adoption of new technologies [63]. The phenomenon of path dependency can be observed in a change of health policy [64], or a business's resistance to a resource reconfiguration process [65]. Path dependency experiences cause people to develop consistent beliefs and reduce the conflicts of belief in cognitive processing. Two cases were found in the literature regarding the cause–effect of path dependency to reinforce the formation of belief consistency. First, people formed an impression of a target as a coherent and unified entity by encoding external information to match their internal boundary conditions [66,67]. Second, the spread of similar experiences in the crowd caused the propensity for herding in the context of information cascades [68]. From most previous studies of review helpfulness (Table 1), the underlying mechanism of review processing is built on the cognitive-information lens. The cognitive-information lens has been widely addressed, such as cognitive processes in information processing or the cognitive systems that connect inputs and outputs [69], cognitive biases in information processing regarding the improvement of information systems [70], review perception according to the construal-level theory [71], cognitive resources invested in review information processing [72], and how cognitive biases of crisis information lead to decision reliance on biased information [73]. According to the cognitive-information lens, cognitive and informational influences should be examined in user-generated persuasion. User-generated persuasion generally affects consumers through attracting attention, reallocating cognitive resources, and evoking affective or emotional responses underlying different levels of cognitive effort in information processing [27], which are considered as *cognitive influences*. *Informational influences* develop in the judgment of the collection of relevant product reviews (or user-generated persuasion) concerning source- and content-based features [17].

The cognitive-information lens was developed to address the embeddedness of new information that can conform to prior beliefs of information processing. Despite the processing of biased or unbiased information, overconfidence in personal knowledge fosters biased judgment. Alternatives may consider that neither biased judgment nor biased information is the right theoretical lens to examine review helpfulness. For example, product type [4,38], reviewer credibility [37], and two-sided persuasion [17] are considered as unbiased judgment or unbiased information in the examination of review helpfulness. However, the conflicts of cognition between consumers and experts [2], the conflicts of informativeness between product popularity and product intangibility [74], the conflicts of review processing between central and peripheral routes [37], the conflicts of emotional influence [75], and the conflicts of influence between informational and normative factors [17], reveal that biased judgement and biased information matter in the evaluation of review helpfulness. To restore psychological comfort, we applied the cognitive-information lens to examine how user-generated content can be congruent with consumption experiences or decision-making contexts. We adopted "belief consistency" to express the psychological state of

cognitive processing, but did not take it as an estimate of initial beliefs or a subjective estimate of intuitions. Belief consistency is built on a preferred path of cognitive processing experiences that connect self-enhancement with self-verification [62], which can enhance continuity and credibility of consistent beliefs, respectively.

**Table 1.** Representative research of review helpfulness.

| Theoretical Foundation | Antecedents | Methodology/Context | Results | Reference |
|---|---|---|---|---|
| A trade-off between perceived costs and perceived benefits of information searching. | Review extremity and review count. Products types: search versus experience products. | Review materials: online reviews of 6 products (3 experience and 3 search) on the Amazon website. Experience products: a music CD, an MP3 player, and a video game. Search products: a digital camera, a cell phone, and a laser printer. | Product type moderates the effect of review extremity on review helpfulness. Extreme ratings are less helpful than moderate ratings in the review evaluation of experience products. Regarding the evaluation of review helpfulness, review count is better for examining search products than examining experience products. | [4] |
| Explanatory and predictive models based on machine learning. | Review subjectivity features and review readability features. Reviewer-related features. | Text mining and sentimental analysis. Review materials: a review set of selected products (e.g., audio and video players, digital cameras, and DVDs). Context: Amazon website and its voting systems. Measures of review helpfulness = [number of helpful votes/number of total votes] for a review. | Reviews that mixed subjective with objective messages are perceived as more helpful. | [76] |
| Dual process theories—elaboration likelihood model, heuristic systematic model. | Central cues: word count and the percentage of negative words. Peripheral cues: rating consistency, reviewer ranking, reviewer real name. Product type: search versus experience, high-price versus low-price. | Web data mining of reviews and reviewers from the 28 product categories on the Amazon website. | Two peripheral cues (e.g., rating consistency and reviewer credibility) and one central cue (e.g., review content) affect the helpfulness of reviews. Central cues are better to judge the review helpfulness of search and high-price products. Peripheral cues are better to judge the review helpfulness of experience and low-price products. | [37] |
| Bach's (1967) helping behavior model. | Source-based review features. Content-based review features. Review helpfulness is a formative construct that is measured in terms of source credibility, content diagnosticity, and vicarious expression. | A 2×2 factorial experiment. Context: An online shopping website uses scenario-based surveys of making purchase decisions under two information types—product features and product reviews from either experts or customers. | Source- and content-based review features determine review helpfulness. Customer-generated reviews are perceived as more helpful than expert-generated reviews. A concrete product review is perceived as more helpful than an abstract one. The interaction between source- and content-based review features can shape review helpfulness. | [2] |
| The emotion-cognition information processing model. | Emotions (e.g., anxiety and anger) embedded in product reviews. Perceived cognitive efforts. | Two lab experiments and one field study using archival data from the Yahoo!Shopping website. | Anxiety-embedded reviews are perceived more helpful than anger-embedded reviews. The effects of negative emotions on review helpfulness are better explained by the beliefs of reviewers' cognitive effort. | [75] |

**Table 1.** *Cont.*

| Theoretical Foundation | Antecedents | Methodology/Context | Results | Reference |
|---|---|---|---|---|
| The search-experience paradigm. The source-content-context model. | Product type: search versus experience. Source factors: reviewer rank, the disclosure of reviewer identity. Context factor: number of reviews for a product. Content factors: review extremity, review depth. | A total of over 28,000 product reviews across 10 product types were collected from Amazon.com (Korea). | Positive determinants of review helpfulness: reviewer reputation, review depth. The number of reviews and the disclosure of reviewer identity have a stronger effect on perceived review helpfulness for experience products. Reviewer reputation, review extremity, and review depth have a stronger effect on perceived review helpfulness for search products. | [38] |
| Dual-process theory | Informational influence: two-sided reviews, source trustworthiness, source credibility, source homophily. Normative influence: e-retailer's, recommendation, service popularity. | Context: Hong Kong International Airport. Sample: Passengers. Materials: Customer reviews about accommodation and restaurants. | Two-sided reviews are perceived as more helpful. Reviews from source expertise are perceived as more helpful. Reviews of service popularity are perceived as more helpful. | [17] |
| The source-content-context model | Source factors: reviewer experience, reviewer expertise. Content factors: review extremity, review inconsistency, review depth. Context factors: product intangibility, product satisfaction, product popularity, product variety. | A total of 14 million review data across 10 product types were collected on Amazon.com. | Positive effects on review helpfulness: review extremity, review depth, reviewer expertise. Negative effects on review helpfulness: review inconsistency, product intangibility, product satisfaction, product popularity, product variety, reviewer experience. Review extremity and review depth determine review helpfulness, depending on product intangibility. | [74] |

### 2.2. Research Model and Hypothesis Development

According to the study of confirmation biases [39], we inferred that people pursue path-dependent experience, i.e., rely more on preferred and habitual routes to reinforce their existing belief systems, to develop consistent rather than inconsistent beliefs of information processing. The approaches to preferred and habitual judgments also exist in complicated economic decisions according to status quo biases [77]. User-generated opinions are filled with biased judgment and/or biased information that may render inconsistent messages. Inconsistent messages easily foster cognitive conflicts in processing new information and thereby increase cognitive costs. For consumers, the formation of belief consistency is a better way to reduce the cognitive costs of review processing. Building on the cognitive-information lens, we developed a research model (Figure 1) to explain how belief consistency affects the helpfulness beliefs of eWOM, and examined the effects of the positive–negative review frame given that cognitive costs can be reduced by using consistent beliefs [69]. We employed the cognitive-information lens to address how the formation of biased judgment (e.g., belief consistency) reinforces the processing of biased (e.g., positive, negative messages) and unbiased (e.g., argument quality) review information. The cognitive-information lens is similar to the elaboration likelihood model (ELM) that relies on cognitive processing and attributes to persuasion [78]. Compared with the ELM which addresses the choices of central and peripheral routes, our cognitive-information lens is better developed to examine the pre-belief-information–post-belief link.

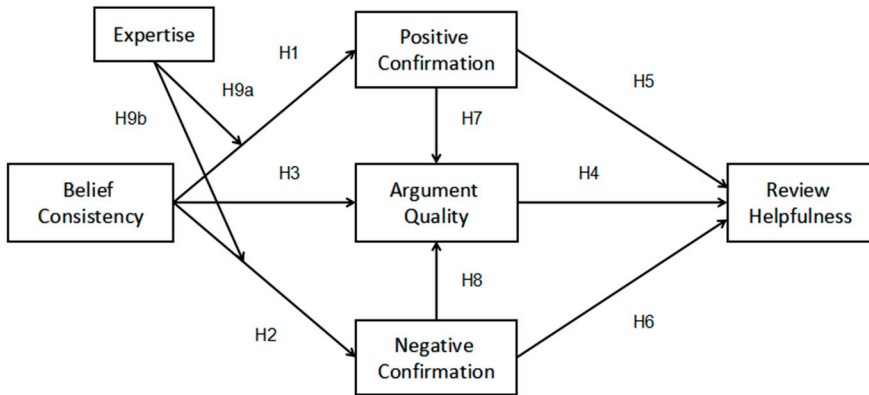

**Figure 1.** Research model.

2.2.1. Belief Consistency

Owing to the limited attention capacity to absorb massive numbers of online reviews [79], consumers are more inclined to search for review information that can conform to their pre-existing beliefs [80]. According to confirmation biases, consumers have a tendency to reinforce their initial beliefs of product information based on prior experiences. The processing of information by relying on prior consistent impressions in mind, likely triggers a person's reasoning with narratives in the belief systems. Consequently, the heuristic reasoning conducted on consistent beliefs dominates the use of inconsistent beliefs in their judgment. Belief consistency can help people avoid the uncertainty of information and increase their confidence in decision-making processes [81]. People that have expectations develop prospective meanings in information processing, namely, *confirming information*, referring to "*information content in messages that is consistent with one's previous held understandings and beliefs*" [82], p. 78. Given that human judgment is constrained by rationality [6], we employed belief consistency to denote the presence of *biased judgment*, in which consistent beliefs instead of inconsistent thoughts are more likely to be recalled in making decisions. Accordingly, consumers with belief consistency in mind are inclined to confirm the expectations of review information underlying self-enhancement of their existing beliefs, or self-enhancement bias. The confirmation that positive (negative) eWOM is more useful than negative (positive) eWOM, namely, *positive* (or *negative*) *confirmation*, has been defined as *positivity* (or *negativity*) *effect* by Yin et al. [32]. We posit that:

**H1:** *Belief consistency increases positive confirmation of review evaluation.*

**H2:** *Belief consistency increases negative confirmation of review evaluation.*

The logic of the argument is crucial to the assessment of argument quality [83]. We consider that the logic of argument is evaluated by the receiver's cognitive reasoning, which is often based on personal experience or the logic–knowledge relationship in mind. Hence, the harmony of cognitive reasoning and relevant evidence appropriately lies in the cognitive-information lens. The connection between consistent beliefs and consistent evidence or information fosters the application of the cognitive-information lens to review judgment. However, the conflicts of user-generated persuasion increases cognitive effort in information processing [84]. Moreover, judgment based on inconsistent beliefs causes a more exhaustive cognitive effort in the evaluation of product reviews. To alleviate cognitive effort, consumers tend to take a habitual path to judgment based on prior consistent beliefs. Belief consistency activates the preference for self-verification over the search for review information that can confirm the narratives of personal needs [85]. Under the influence of belief consistency, consumers focus on consistent arguments and thereby perceive a strong argument quality of the review content that underlies a feedback loop of post-belief reinforcement [62]. We posit that:

**H3:** *Belief consistency increases the argument quality of reviews.*

### 2.2.2. Argument Quality

Argument quality refers to the specific quality that makes arguments about user-generated content more persuasive [78]. Argument quality of eWOM is measured in terms of the perceptions of review persuasion [86], which is not examined in positive–negative asymmetry. The adoption of argument quality can provide alternative explanations of helpfulness beliefs other than positive–negative asymmetry. Argument quality is well-developed and widely examined in the literature [86–89], which is also crucial to the evaluation of review helpfulness because of two supportive reasons. First, the review messages which have argument quality easily attract recipients' attention to the persuasive information [90]. Second, argument quality is a central cue for review quality [86], which needs more cognitive effort in review judgment. Argument quality reflects the beliefs of review information. Regardless of the level of personal expertise, the argument quality of user-generated persuasion makes a review more credible [86]. Consumers that perceive the stronger argument quality of an online review are more likely to judge the review as helpful. We, thus, posit that:

**H4:** *Argument quality positively affects review helpfulness.*

### 2.2.3. Positive and Negative Confirmation

Given that cognitive load on human judgement does matter under uncertainty, people adapt to the availability heuristic of consistent beliefs or highly familiar thoughts to develop their mental process, i.e., the subconscious process of a current search set instead of perfect rationality, in making their judgment [6,8]. The positive–negative anticipatory emotions that reflect experience information matter in social communication [91]. Obviously, the positive–negative review frame reflects reviewers' experience information and self-confirmed persuasion [92]. Reviewers' emotions holding experience information, such as anxiety and anger in eWOM persuasion, shape a recipient's helpfulness perception [71]. We take the positive–negative frame as an availability heuristic of information processing for consumers to seek desirable, and drop undesirable, sentimental messages, which is better in reducing cognitive load in the evaluation of massive numbers of product reviews. Positive confirmation of review evaluations is not only linked to product quality, but is also used to exhibit the reputation signal in markets [93]. In contrast, consumers often perceive negative reviews as diagnostic information, and thereby perceive negative confirmation of product reviews as a signal of warning [37]. A positive–negative review frame that produces either positive or negative confirmation easily draws comparisons between intended and unintended influences—whether positive or negative eWOM is more useful. In sum, we posit that:

**H5:** *Positive confirmation of review evaluation increases review helpfulness.*

**H6:** *Negative confirmation of review evaluation increases review helpfulness.*

Argument quality is also defined as the perceived quality of arguments (or persuasive messages) in an online review to achieve informational influence [42]. Argument quality is similar to review impression in terms of informational influence. The debate about whether positive or negative reviews are more helpful was examined in the literature [22,94]. Yin et al. [32] considered positive–negative asymmetry as a consequence of confirmation biases. From the perspective of the positive–negative frame, extreme eWOM is more attractive than moderate eWOM in the context of review persuasion [74]. Obviously, reviewers are more likely to report either higher (positive) or lower (negative) ratings rather than average ratings in the spread of consumption experience [95]. The processing of product reviews may foster the desire for affective review content, i.e., the phenomenon of consumers seeking the confirmation of either positive or negative information. Consumers that take affect-confirmation processes, i.e., confirmation of positive and negative affect influences, into their belief systems [96], are more likely to take the confirmation of positive or negative reviews as an affective signal of review content or useful experience information,

and thereby consider the reviews as more credible in persuasion. The opposite view—how argument quality affects perceived usefulness—is contingent on the choice of a central or peripheral route, because a person's elaboration may change over time [41]. Relative weighting of positive or negative review information reinforces the readers' impression of the reviews [97]. Similarly, we consider that positive or negative confirmation of review information reinforces the argument quality of reviews. We, thus, posit that:

**H7:** *Positive confirmation of review evaluation increases the argument quality of reviews.*

**H8:** *Negative confirmation of review evaluation increases the argument quality of reviews.*

### 2.2.4. Expertise

Consumers with more expertise of some specific products are more likely to evaluate product quality by themselves; alternatively, they may also read product reviews only because of a desire to seek shared views or common feelings about the consumption experience. In contrast, consumers with less expertise are more likely to read product reviews before purchasing. Owing to different levels of expertise, consumers form distinct beliefs toward user-generated persuasion. Consumers with high (low) levels of expertise indicate that they have a strong (weak) ability to judge user-generated content, such as product reviews. Strong ability increases the confidence levels of cognitive processing, leading to favored judgments. From confidence in initial beliefs to confidence in existing beliefs, cognitive processes, i.e., connected beliefs of a target, produce more consistent beliefs without effortful examination against belief inconsistency [31]. Prior experiences often stick in belief systems to increase consumers' self-enhancement, and against a judgment based on inconsistent information [32]. Personal expertise reflects a person's cognitive processing abilities [42], which may foster the use of consistent beliefs in the processing of positive or negative review information.

The separation between consumers' and critics' product reviews is crucial to the examination of review credibility [98]. People often consider the expertise level of critics to be greater than that of consumers, and thereby enlarge the credibility of critics' reviews. Expertise may play a role, as the ability of review evaluation, or the motivation to seek eWOM, depends on the communication processes between viewers and reviewers. Despite the possibility that the claimed expertise of people might inflate their self-efficacy beliefs [99], we considered the effect of overconfidence behind personal expertise. In this study we set the contexts of exposure to eWOM, rather than examining the effects of eWOM communication. We inferred that *personal experience* reinforces the connection between existing beliefs and prior beliefs via path-dependent judgment [62], leading to more confidence in consistent beliefs and the confirmation of positive or negative reviews. We, thus, posit that:

**H9a:** *Expertise increases the effect of belief consistency on positive confirmation of eWOM.*

**H9b:** *Expertise increases the effect of belief consistency on negative confirmation of eWOM.*

### 3. Methodology

#### 3.1. The Instrument

Dropping other potential causes (e.g., selective information searching and sequential presentation) and direct effects of confirmation bias [54,55], we focused on the effect of belief consistency on review judgment. We examined "belief consistency" in terms of the consistency between existing beliefs of selected eWOM and associated knowledge, prior impressions, and the reviews of mobile phones that were built on personal belief systems, using a scenario-based questionnaire survey. The measurement of belief consistency depended on the context that the participants were asked to match the review set to their memories in the recollection process. The selected four reviews were adopted from third-party sites—external eWOM—and examined by text-mining tools according to word counting and sentimental strength. Positive or negative confirmation, i.e., positive–negative asymmetry, was measured in terms of usefulness, informativeness, and influence, in the

comparisons between positive and negative eWOM. Argument quality was changed in wording and measured using the scales of Cheung et al. [86]. Expertise was measured in terms of informative understanding, professional experience, and sufficient knowledge about the eWOM of mobile phones. Except for the helpfulness scores, we operationalized the measurement items of the other constructs by either modifying the scales adopted from previous studies or developing new measures to meet the survey context (Appendix A). Respondents were asked to judge review helpfulness in terms of a score from 1 to 10, which might reduce concern on multicollinearity in the measurement. Other measurement items were anchored using the five-point Likert scale. A pre-test was conducted to improve the content validity of the measurement items before the formal survey.

*3.2. Data Collection*

Apple is a very valuable company and also a strong brand that might foster the building of strong prior impressions in people's minds. Unavoidably, the selection of iPhone associated with eWOM might fall in the situation of potential selection biases, just as for the selection of other products. The selection of brands or reviews was a key concern in this study because the phenomenon of selection biases might exist. To achieve a balanced evaluation, the review set had two reviews that supported Apple, and two reviews that were against it.

The balance and sequence of review information are considered to shape the formation of impressions and activate relative recall from memories [97]. The balance of words of the positive–negative frames and the sequence of prior impressions were unlikely to be completely controlled in the survey. Random sampling was a potential solution for fixing the two barriers. However, personal choices in real life do not completely reflect random sampling. Respondents were asked to read the mixed eWOM of mobile phones in the scenario-based questionnaire survey. We employed sentimental messages of mobile phones regarding a comparison between Apple iPhone and other brands, which were evaluated to confirm as positive or negative reviews according to the respondents. We collected mobile phone reviews that had been posted on Chinese websites within one month prior to our research work.

Using a web-based system control, we guided each respondent to first indicate their personal expertise regarding mobile phones, and then to read the review messages for product evaluations and purchase decisions before replying to the questionnaire. To reduce the contrast effect of the survey, we presented the same sequences of reviews. We surveyed for target users that had read eWOM of mobile phones before the survey. This procedure ensured that all respondents recalled prior beliefs of mobile phone eWOMs in their memories before replying to the questionnaire. Given that respondents may search for more eWOM of other mobile phones outside the web-based system, the review set was used to reduce external noise in the survey. We posted the questionnaire on an online survey website to collect the targeted sample. All respondents were asked to report their helpfulness scores for the four reviews using a spectrum scale ranging from totally unhelpful to totally helpful, and an overall helpfulness score of the reviews. During a five day period, we obtained 334 (male = 189, female = 145) completed questionnaires. Overall, our samples were drawn from a large database of local consumers and, thus, might represent the population of mobile phone users.

## 4. Results

*4.1. Reliability, Validity, and Common Method Variance*

Except for one item construct—review helpfulness—we examined the reliability and validity of the five constructs that were developed to test the hypotheses. The factor analysis (Table 2) indicated that all factor loadings exceeded the corresponding cross-loadings and the 0.6 threshold. From Table 3, both Cronbach's alpha and composite reliability of the five constructs significantly exceeded the 0.70 threshold [100], achieving internal consistency. The average variance extracted (AVE), i.e., the square score of the diagonal

element, exceeded the threshold value of 0.5 (Table 3), showing acceptable convergent validity [101]. From the correlation analysis matrix (Table 3), the square roots of the AVE in the diagonal were greater than the off-diagonal construct correlations, achieving discriminant validity [101]. The common method variance was examined to reduce potential concerns on the questionnaire method (Appendix B). We examined the variance inflation factor (VIF) of the measurement items; the VIF scores were less than the cut-off of 5 [102], suggesting that the multicollinearity problem was not a major concern in this study.

**Table 2.** Factor loadings and cross-loadings of measurement items.

| Item | AQ | E | NC | PC | BC |
|------|------|------|------|------|------|
| AQ1 | **0.817** | 0.112 | 0.238 | 0.237 | 0.101 |
| AQ2 | **0.827** | 0.078 | 0.194 | 0.204 | 0.146 |
| AQ3 | **0.827** | 0.073 | 0.202 | 0.209 | 0.227 |
| AQ4 | **0.766** | 0.086 | 0.139 | 0.216 | 0.207 |
| E1 | 0.140 | **0.874** | 0.005 | 0.121 | 0.119 |
| E2 | 0.037 | **0.933** | 0.027 | 0.141 | 0.034 |
| E3 | 0.094 | **0.933** | 0.027 | 0.127 | 0.028 |
| NC1 | 0.220 | 0.036 | **0.890** | 0.109 | 0.066 |
| NC2 | 0.181 | 0.020 | **0.895** | 0.106 | 0.143 |
| NC3 | 0.203 | 0.007 | **0.889** | 0.119 | 0.130 |
| PC1 | 0.311 | 0.119 | 0.121 | **0.839** | 0.153 |
| PC2 | 0.208 | 0.154 | 0.143 | **0.884** | 0.094 |
| PC3 | 0.246 | 0.183 | 0.100 | **0.871** | 0.142 |
| BC2 | 0.449 | 0.267 | 0.182 | 0.226 | **0.621** |
| BC3 | 0.329 | 0.033 | 0.221 | 0.203 | **0.823** |
| Eigenvalue | 3.286 | 2.682 | 2.667 | 2.612 | 1.298 |
| Cumulative Variance (%) | 21.906 | 39.787 | 57.564 | 74.978 | 83.633 |

Principle component analysis; rotation: Varimax; KMO = 0.876, $\chi^2$ = 3847.465, *df* = 105. Bold: Indicates significant outputs that exceed the threshold of 0.6.

**Table 3.** Tests of reliability and validity.

| Construct | Mean | SD | AQ | E | NC | PC | BC | Cronbach Alpha | Composite Reliability |
|-----------|------|------|------|------|------|------|------|------|------|
| AQ | 3.32 | 0.77 | **0.810** | | | | | 0.90 | 0.93 |
| E | 3.16 | 0.88 | 0.327 | **0.914** | | | | 0.92 | 0.93 |
| NC | 3.41 | 0.80 | 0.409 | 0.134 | **0.891** | | | 0.92 | 0.94 |
| PC | 3.49 | 0.78 | 0.532 | 0.368 | 0.299 | **0.865** | | 0.92 | 0.94 |
| BC | 3.49 | 0.66 | 0.617 | 0.373 | 0.387 | 0.460 | **0.729** | 0.74 | 0.84 |
| RH | 6.88 | 1.84 | 0.467 | 0.228 | 0.234 | 0.383 | 0.367 | none | none |

Diagonal elements represent the squared roots of the average variance extracted (AVE) of the constructs, while the other matrix elements represent the inter-construct correlations. Bold: Indicates significant outputs that exceed the threshold of 0.6.

Before testing the hypotheses, we needed to first examine the measurement model, then secondly examine the structural model [103]. We examined the measurement model using the sample data. The indexes of the measurement model indicated a good fit (Table 4). Hence, we confirmed the results of factor analysis, that the items of the five constructs were suitable for testing the structural model. Subsequently, we examined the two structural models, without and with moderation, and all goodness-of-fit indices achieved acceptance levels (Table 4). Hence, the two structural models were adequate to examine the hypothesized cause–effect using the sample data.

**Table 4.** Tests of the measurement model and the structural model.

| Goodness-of-Fit Indexes | Recommended Threshold | Measurement Model | Structural Model | Moderation (Expertise) |
|---|---|---|---|---|
| $\chi2/df$ | ≤3.00 | 1.781 | 2.259 | 1.296 |
| GFI | ≥0.90 | 0.948 | 0.995 | 0.992 |
| AGFI | ≥0.80 | 0.923 | 0.960 | 0.969 |
| NFI | ≥0.90 | 0.964 | 0.992 | 0.986 |
| CFI | ≥0.95 | 0.984 | 0.995 | 0.997 |
| SRMR | ≥0.05 | 0.033 | 0.025 | 0.023 |
| RMSEA | ≥0.08 | 0.048 | 0.061 | 0.030 |

Recommended threshold: Bagozzi and Yi [104]; Wheaton et al. [105]; Hair et al. [106].

*4.2. Hypothesis Testing*

Our empirical results (Figure 2) indicated that belief consistency positively and significantly affected positive and negative confirmation and argument quality, supporting hypotheses H1, H2, and H3. Consumers with consistent beliefs of eWOM were more likely to perceive positive or negative eWOM as useful, and also considered eWOM as more persuasive. Argument quality and positive confirmation, rather than negative confirmation, significantly affected review helpfulness, supporting hypotheses H4 and H5 rather than hypothesis H6. Consumers considered persuasive or useful positive eWOM as the approach to favorable outcomes and, thus, helpful for making the purchase decision. Negative confirmation reviews appeared to be an avoidance of unfavorable outcomes, and, thus, less helpful for judging the targeted products. As hypotheses H7 and H8 predicted, positive confirmation and negative confirmation were two significant determinants of argument quality. Useful eWOM, either positive or negative, is considered as persuasive for consumers. The effect of belief consistency on positive confirmation was not contingent on personal expertise, and, thus, hypothesis H9a was not supported. In contrast, the effect of belief consistency on negative confirmation was stronger for respondents with higher levels of expertise, supporting hypothesis H9b. Consumers were more likely to apply personal expertise to the usefulness evaluation of negative reviews, probably because more ability is required in the evaluation of diagnostic information. From the ad hoc test, personal expertise was a positive determinant of positive confirmation of review evaluation. Moreover, the effect of belief consistency on argument quality was not contingent on personal expertise in our ad hoc test. In summary, the theoretical model accounted for 26.5% of variance in perceived review helpfulness.

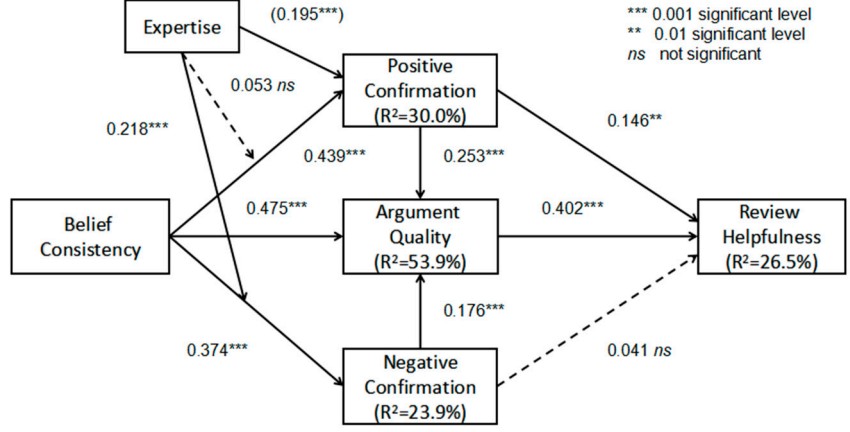

**Figure 2.** Empirical results (moderation analysis).

## 5. Discussion

From the sample statistics, nearly one third of respondents, 32.1%, reported that none of the four reviews were helpful in making their purchase decisions. Moreover, about 64.6%

of respondents reported that one or two reviews were helpful, and just 3.3% of respondents reported that three or four reviews were helpful. However, the four online reviews we selected from local websites had received a large number of "like" clicks and exhibited high ranking during the survey period. Given the conflicts of preference for consumers, the views of judging the selected reviews as helpful were totally an illusion of self-verification.

In case of the positive–negative review frame, judgment based on review content was more complex than the frame itself. We will answer the first question: that *the positive-negative asymmetry* (*e.g., confirming positive rather than negative information effect*) is significant to the evaluation of review helpfulness. Interestingly, consistent beliefs of positive information cause people to perceive positive eWOM as helpful, while consistent beliefs of negative information do not cause people to perceive negative eWOM as helpful. Compared with negative review information, positive review information is likely to produce a stronger influence, underlying the overconfidence in consistent beliefs [107]. Conflicts of confirming information are inevitable to constrain the positive–negative frame of user-generated content, and ultimately arouse a debate about the relative weighting of review information in review evaluation. Our explanation is that consumers prefer an approach towards the desired outcomes (e.g., positive confirmation) rather than an avoidance of undesired outcomes (e.g., negative confirmation) when seeking helpful reviews to achieve better purchase outcomes, showing that the conflicts of the positive–negative review frame should be examined in user-generated persuasion. The confirming information, either positive or negative, produces the intended influence to increase the argument quality of user-generated content, leading to self-verified confidence. We now answer the second question about how belief consistency determines the helpfulness beliefs of eWOM. Our empirical results basically supported the view that consistent beliefs fostered the confirmation of positive reviews to shape helpfulness beliefs. From consistent beliefs to the confirmation of negative reviews, the perceptual connection between helpfulness beliefs and purchase decisions was probably weakened. The third question regarding whether personal expertise fosters or inhibits the approach to rely more on consistent beliefs in review judgment, is a *yes-and-no* answer. For high levels of expertise, belief consistency can reinforce negative-confirming information, but does not reinforce positive-confirming information. The post hoc test indicated that belief consistency and expertise were mutually independent in shaping positive-confirming information. A possible explanation is that the processing of positive reviews relies more on either consistent beliefs or useful experience information [19], while the processing of negative reviews or diagnostic information relies more on the combination of consistent beliefs and personal expertise [20].

*5.1. Implications for Theory*

The inference regarding how people's belief systems distort the valuation of information is derived from the effect of confirmation biases [39]. The proposition to consider that helpfulness beliefs of product reviews stems from people's confirmation biases is totally exaggerated and not the purpose of this study. However, the processing of user-generated persuasion is probably driven by biased information and/or biased judgment, given that subjective or emotional messages are spread in the persuasion [76]. Building on two assumptions, we took the combination of belief consistency and positive–negative asymmetry as the cognitive-information lens to examine the connection between biased judgment and biased information in the context of user-generated persuasion, which has been little examined in the literature. We addressed whether the approach to biased judgment or biased information has a significant effect on the evaluation of review helpfulness, which extends this knowledge in the literature.

In this study, we posed one thoughtful question: Whether perceived review helpfulness is a belief of information, or an illusion of overconfidence? It is likely that the former is constrained by biased information, while the latter is constrained by biased judgment [108]. To simplify the solution to the question, we addressed the approaches to foster the development of biased information and biased judgment in user-generated

persuasion. Biased information and biased judgment are more likely to be embedded in wisdom-of-crowds (e.g., eWOM), and less likely to survive in scientific research. According to the empirical results of this study, the conflicts of influence between positive and negative confirmation indicated that perceived review helpfulness is a belief of information and is constrained by the positive–negative review frame. Without using personal expertise, respondents' consistent beliefs are significant in the confirmation of positive reviews as useful and thereby the perception of the review information as helpful, which is totally an illusion of overconfidence and constrained by belief consistency. Prior studies considered that the helpfulness of product reviews was judged on unbiased information or unbiased judgment [2,4,109,110]. Yet, we considered "argument quality" as one type of unbiased information that can increase helpfulness beliefs. The first theoretical implication is that perceived review helpfulness is both a belief of information and an illusion of overconfidence.

The presence of confirmation biases mostly stems from overconfidence in prior beliefs [31], which can be used to explain why belief consistency exists, but not to examine how belief consistency works, as examined in the study. Given that in the recall of prior beliefs, personal expertise is hypothesized to reinforce the effect of belief consistency on the positive and negative confirmation of eWOM evaluations. The mixed empirical findings that belief consistency increases negative rather than positive judgement of review information created with high levels of expertise, indicates that belief consistency may work in different conditions, which needs reasonable explanation. We provided the explanation that consistent beliefs are reinforced in cognitive processing which weakens the use of personal expertise in the confirmation of positive reviews. The interaction between information cues and cognitive effort was hypothesized to determine the route to information evaluation [78] or review helpfulness [111]. In case of intended persuasion via positive and negative review information, the conflicts of cognitive influence do not support the view that biased judgment is attributable to personal expertise. The second implication is whether personal expertise reinforces the effect of belief consistency depends on positive–negative asymmetry. In this study, we have no evidence to take personal expertise as a fostering condition of belief consistency in product review evaluations.

Overall, the empirical results of this study supported the connection between pre-belief (e.g., consistent beliefs) and post-belief (e.g., helpfulness beliefs) that underlies the positive–negative review frame. It is reasonable to apply the pre-belief-information–post-belief link to examine how biased judgment and biased information might distort review judgment. Accordingly, we inferred a reinforcement-influencing cycle, i.e., more consistent beliefs lead to the collection of more consistent or one-sided information, and the collection of more consistent or one-sided information leads to the formation of more consistent beliefs. The third implication is that a reinforcement-influencing cycle is inferred from using the cognitive-information lens to examine the positivity/negativity confirmation of user-generated persuasion that is consistent with the view of positive affective value [52].

### 5.2. Implications for Practice

The ongoing phenomenon of many fake online reviews targeting consumers to manipulate their belief systems is bothersome [112–115]. The intended infection of eWOM invokes our concern that people are unaware of the unexpected effect of consistent beliefs or are even inclined to rely on a habitual way of reviewing judgment based on their beliefs. In the marketing of online products, eWOM is a signal of emotional infection or an outcome of marketing promotion via the spread of user-generated persuasion [116]. In practice, many platform businesses (e.g., TripAdvisor, Yelp) already know how to adapt user-generated persuasion to reinforce consistent judgment on helpful votes, rather than searching for evidence of review inconsistency [117,118].

People are suffering from biased information mixed with opinions and thoughts that is embedded in user-generated persuasion. The use of WOM or eWOM instead of price information to estimate product quality or customer choice is far more than exhausting to

make a judgment. For recipients, the almost free information on user-generated content may incite them to neglect the hidden costs or cognitive efforts. The barrier to consumer choices is obvious, in that firms do not pay very much for the cost of eWOM evaluation. In contrast, consumers pay the cost of eWOM evaluations, such as in the cognitive effort invested in processing biased information, or biased judgment in the widespread presence of fake reviews [119–121]. To reduce cognitive overload, people tend to rely more on their habitual patterns of judgment, in which consistent beliefs are recalled more frequently than inconsistent beliefs in judgment. Belief consistency, instead of inconsistent thinking, becomes the availability heuristic to render biased judgment. On seeing many five-star product ratings on e-commerce websites, i.e., the availability heuristic of more consistent information, consumers may perceive those product reviews as being more helpful. Over-expected volumes of five-star product ratings might manifest in the presence of biased judgment or biased information [95]. The helpfulness scores seem to exhibit the "collective intelligence" of product evaluations, but the dispersion of average ratings is exposed to produce intended influences on consumers [122]. The first implication concerns the beliefs of collective wisdom and the messages in user-generated persuasion, which become the sources of biased judgment and biased information to mislead single-minded consumers.

This study is not an approach to address an accurate estimation formula or an answer to the evaluation of review helpfulness. Normally, consumers often seek specific review features to judge the helpfulness of an eWOM. However, incomplete information about specific features of eWOM, conflicts of interest in product evaluation, and unstable preferences over time, might reinforce the habitual use of belief systems to reduce a person's cognitive effort. According to the wisdom-of-crowds [15], eWOM is believed to create a lower score of the average error of individuals. However, belief consistency may create a higher score of an individual's average error. Treating the chance of biased information on user-generated persuasion as controllable over review processing seems to be an illusion of overconfidence [108]. We consider that belief consistency enlarges the error of review judgment on an individual's average error. The second implication concerns how to develop more useful belief systems or narratives about user-generated content—the avoidance of biased judgment is better than the prevention of biased information, for reducing the average error of product review evaluations. More review features are useful for seeking helpful eWOM, but they are easily dropped in applying the least effort strategy to eWOM evaluation.

## 6. Conclusions and Limitations

Without holding consistent beliefs in mind, it is an effort for consumers to examine product reviews that are mixed with positive and negative information. The usefulness beliefs of positive or negative eWOM were hypothesized to increase helpfulness beliefs. Our empirical results did not completely support the hypotheses. The useful–helpful link did work in the evaluation of positive reviews, but was insignificant in the evaluation of negative reviews. However, the useful–quality link was verified in the evaluation of either positive or negative eWOM. The conflicts of the useful–helpful link may reflect the effects of biased information on the evaluation of user-generated persuasion. A positive eWOM was perceived as useful in the condition of consistent beliefs regardless of the levels of personal expertise. A negative eWOM was perceived as useful in the condition of consistent beliefs and personal expertise. The conflicts of cognitive processing, such as the conflicts of applying personal expertise and consistent beliefs to judge the usefulness of positive and negative eWOM in the study, can provide an approach to examine the boundary condition of the cognitive-information lens.

The limitations of this study should be considered in future work. The selection of framing in user-generated content was the first limitation of this study. This limitation is an incentive to encourage further study using different frames of review information. The second limitation was the generalizability of mobile phone eWOM. The four reviews cannot represent all other review information. The selection of only a few online reviews also falls into the concern of selection biases. We assumed the existence of belief consistency

and did not address an approach to achieve consistent beliefs, which is a self-verification measurement, and thereby considered as the third limitation of the theoretical foundation. The fourth limitation was the reinforcement-influencing cycle, that should be examined in the study of other cognitive and informational factors. From the little explained variance in review helpfulness, we acknowledge the fifth limitation as being the design of helpfulness scores.

**Author Contributions:** Conceptualization, H.-P.S., K.-h.L. and T.C.E.C.; methodology, H.-P.S.; validation, H.-P.S.; formal analysis, H.-P.S.; investigation, H.-P.S.; data curation, H.-P.S.; writing—original draft preparation, H.-P.S.; writing—review and editing, K.-h.L. and T.C.E.C.; funding acquisition, H.-P.S. All authors have read and agreed to the published version of the manuscript.

**Funding:** This research was funded by [Social Science Foundation of Fujian Province] grant number [FJ2021B166] And The APC was funded by [FJ2021B166].

**Institutional Review Board Statement:** Not applicable.

**Informed Consent Statement:** Not applicable.

**Data Availability Statement:** The data used in this study are available on request from the corresponding author.

**Acknowledgments:** The authors thank the two anonymous referees for their valuable comments and suggestions on earlier versions of this paper.

**Conflicts of Interest:** The authors declare no conflict of interest.

## Appendix A. The Measurement Items

Suppose that you are thinking of purchasing a new mobile phone. Interested in Apple's iPhone, you set out to search online reviews from experienced users in order to seek helpful advice or recommendations. We provide the following four online reviews of iPhone from third-party websites for your reference and further consideration for judgment (Table A1). Please also reply to the questionnaire survey after reading the selected online reviews.

**Table A1.** Selected online reviews of mobile phones.

| No. | Sentimental Messages |
|---|---|
| 1 | "iPhone is so expensive and iOS is a close system. While Android is not difficult to use, and it has more freedom of options and no function difference compared with iPhone. Why so many people like iPhone, just because of lost in fashion?" |
| 2 | "I use Sony (mobile phone), whereas my younger brother uses iPhone. I feel the choice of mobile phones depends on your budget. I will choose Sony Xperia Active Sport because this type of mobile phone is unique with small panel. I choose it because of my personal needs in sport, and it provides anti-water, anti-dust, and anti-friction functions. Moreover, it supports with sport-used sensors, and also provides step-counting and music functions. I have no special needs in videos and games. The price indeed suits my personal needs. I never take iPhone into account." |
| 3 | "I feel iPhone is more valuable because it provides effective and efficient APPs." |
| 4 | "iPhone is easy to use, and so fashion and attractive. But its APPs are not so cheap." |

Belief Consistency of Product Reviews (BC) (Source: self-developed)
BC1 The reviews are inconsistent with my prior knowledge (dropped).
BC2 The reviews are consistent with my prior impressions.
BC3 The reviews are similar to other reviews in my mind.
Positive Confirmation (PC) (Source: self-developed)
After reading the review set,
PC1 I feel the positive reviews are more useful than the negative reviews for judgment.

PC2 I feel the positive reviews are more informative than the negative reviews for judgment.

PC3 I feel the positive reviews are more influential than the negative reviews for judgment.

Negative Confirmation (NC) (Source: self-developed)

After reading the review set,

NC1 I feel the negative reviews are more useful than the positive reviews for judgment.

NC2 I feel the negative reviews are more informative than the positive reviews for judgment.

NC3 I feel the negative reviews are more influential than the positive reviews for judgment.

Argument Quality (AQ) (Source: [86])

AQ1 The review arguments are convincing.

AQ2 The review arguments are strong.

AQ3 The review arguments are persuasive.

AQ4 The review arguments are good.

Review Helpfulness (RH)

Please circle a score to indicate your judgment of the review set about its helpfulness.

Scores: (0—1—2—3—4—5—6—7—8—9—10)

Expertise (E) (Source: Modifying the scale from Cheung et al. [86])

E1 I have informative understanding of the review issues of mobile phones.

E2 I have professional experience in the review issues of mobile phones.

E3 I have sufficient knowledge on the review issues of mobile phones.

## Appendix B. Common Method Variance

We examined common method variance (CMV) using a post hoc procedure to test the self-reported data. First, according to Harmon's one-factor test, factor analysis indicated that the largest variance explained by one factor, i.e., the "argument quality" factor, was rightly under 22% (Table 2), which did not explain the majority of the variance in the exploratory study [123]. Second, the test of inter-construct correlations (Table 3) indicated that the highest correlation (0.617) between the research constructs (BC and AQ) was far below the threshold of 0.90 [124]. Third, we examined the correlation between a marker variable, i.e., a theoretically unrelated variable (e.g., education in years), and the constructs of this study [125]. The average correlation coefficient of the marker variable with other constructs was small ($-0.021$). In summary, we considered that CMV was not a major concern in the study.

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
