# Peer review of "Complied by Belief Consistency: The Cognitive-Information Lens of User-Generated Persuasion"

_jtaer, doi:10.3390/jtaer18010020_

Round 1

Reviewer 1 Report (Previous Reviewer 1)

After I read the manuscript, it’s already fine.

But in order for the manuscript good structure still needs to revise the follow my suggestion below:

Page 4, R 183 and page 8, R267

“3. Theoretical Foundation and Hypothesis Development” ( please put this sentence a  number 2, and then put this 2, Then put this “ The Cognitive-Information Lens” as the sub number 2.1 then following with “ 2.2 Belief Consistency  then .others

3. Methodology

Then……

Good luck

2023/1/18

Author Response

Thank you so much for the suggestions.

Reviewer 2 Report (Previous Reviewer 2)

In the previous review was requested major revision, but as we may observe the demands were not revised or improved all (only 1 of 3 were improved and revised).

1.- not fulfilled

Again, in the Introduction is missing the presentation of the article content- each chapter must have a short presentation and also what is brought new, in making this research.

2.- not fulfilled

The study used many sources very old (1957, 1986, 1989, 2001-2011)- only one source is from 2022- thus, it is imposed to be used at least another 15 sources from 2022 from high ranked journals to be appropriate for publication. As we observe, there were added only 4 sources from 2022, so, old added sources (1972 for example), must be replaced, or other new sources added.

3.- fulfilled

Research hypothesis were better explained and added that they were confirmed individually and must be used more results (data).

So, the authors must reconsider after major revision (control missing in some experiments).

Author Response

Thank you so much for the comments.

Round 2

Reviewer 2 Report (Previous Reviewer 2)

Now, all the improvement requirements of the article have been fulfilled, therefore, we propose for publication the article in the present form.

Congratulations!

This manuscript is a resubmission of an earlier submission. The following is a list of the peer review reports and author responses from that submission.

Round 1

Reviewer 1 Report

Comments and suggestion to Authors

A.     Abstract

The abstract needs to rewrite, I couldn’t understand what the author want to explain here.

Suggestion:

1.      When you write abstract, see the problem you want to solve, variables of research model, data, and purpose of study, and the results.

2.      Read more the good article writing.

B.     Key word

Put only necessary variables that related to you model.

C.     Introduction

The introduction is not clear, explained a lot unnecessary unrelated research problem.

Suggestion: See your model, then explain the problem of previous study that related to your present study. and how you solve the problem with your present study, then explain your present study that will help to solve the problem (you need to see your research model)

The research Question below look didn’t matches your research model.

RQ2: Whether personal expertise fosters consumers to rely more on consistent beliefs in 164 review judgment? 165

RQ3: How biased judgment and biased information are interlocked to shape the helpfulness beliefs of eWOM?

These made the purpose of study didn’t clear.

D.    Literature review

1.     The text body in (2. Review Judgment on Consistent Beliefs) can put Literature review but the explanation needs to rewrite that could related to you research model.

2.     When you write the literature review you need to see your variable in you model that will explain is the theoretical background of the present study and the problem related to previous study.

3.     Literature review needs to rewrite. Such as (Table 1. Representative research of review helpfulness) I don’t clear, and didn’t see explanation in related in (2. Review Judgment on Consistent Beliefs)

4.     This text body in (3. Theoretical Foundation and Hypothesis Development) the explanation didn’t clear and didn’t so relate with the model fig 1.

5.     Hypothesis Development H7, H8, H9, H10 look not reasonable needs read more other article.

E.     Methodology

Methodology lots still need to rewrite. Real not clear and hard to understand.

F.     Results

Still many problem – not clear explanation which the most important results of Reliability, Validity, and Common Method Variance.

For example: Table 1, both Cronbach’s alpha and composite reliability of each construct significantly exceed the 0.70 threshold, it’s?

Table 3. Tests of reliability and validity.? Difference in body text

Table 4. Tests of the measurement model and the structural model. I couldn’t understand with this not enough explanation.

G.    And the following also still need to rewrite (6. Discussion, 7. Conclusions and Limitations)

Note: This study still needs to spend a lot time to rewrite and read good others article enable to improve and to publish this article.

Good luck

11/1/2022

Reviewer 2 Report

The topic of the study is interesting, but the article needs improvements in order to be published. First of all, the Introduction is missing the presentation of article content- each chapter must have a short presentation and also what is brought new, in making this research. Research hypothesis must be more explained if they are confirmed or not (the authors used the confirmation in block of research hypothesis)- they have to be confirmed individually and must be used more results (data). And more important, the study used many sources very old (1957, 1986, 1989, 2001-2011)- only one source is from 2022- thus, it is imposed to be used at least another 15 sources from 2022 from high ranked journals to be appropriate for publication. So, the authors must reconsider after major revision (control missing in some experiments).